# Distinctive genes and signaling pathways associated with type 2 diabetes-related periodontitis: Preliminary study

**Poliana Mendes Duarte** [1]<sup>☉</sup>*, **Bruno César de Vasconcelos Gurgel**[2], **Tamires Szeremeske Miranda**[3], **Juliana Sardenberg**[1], **Tongjun Gu**[4], **Ikramuddin Aukhil**[5]<sup>☉</sup>*

**1** Department of Periodontology, University of Florida College of Dentistry, Gainesville, FL, United States of America, **2** Department of Dentistry, Federal University of Rio Grande do Norte, Natal, RN, Brazil, **3** Department of Dentistry, São Judas Tadeu University, São Paulo, SP, Brazil, **4** ICBR Bioinformatics, University of Florida, Gainesville, FL, United States of America, **5** ECU School of Dental Medicine, East Carolina University, Greenville, NC, United States of America

☉ These authors contributed equally to this work.
* Aukhili23@ecu.edu (IA); pmendesduarte@dental.ufl.edu (PMD)

**Data Availability Statement:** All relevant data are within the manuscript and its Supporting Information files.

## Abstract

The biological mechanisms underlying the pathogenesis of type 2 diabetes (T2DM)-related periodontitis remain unclear. This cross-sectional study evaluated the distinctive transcriptomic changes between tissues with periodontal health and with periodontitis in patients with T2DM. In this cross-sectional study, whole transcriptome sequencing was performed on gingival biopsies from non-periodontitis and periodontitis tissues from non-diabetic and diabetic patients. A differentially expressed gene (DEG) analysis and Ingenuity Pathway Analysis (IPA) assessed the genes and signaling pathways associated with T2DM-related periodontitis. Immunohistochemistry was performed to validate selected DEGs possibly involved in T2DM-related periodontitis. Four hundred and twenty and one thousand five hundred and sixty-three DEGs (fold change $\geq$ 2) were uniquely identified in the diseased tissues of non-diabetic and diabetic patients, respectively. The IPA predicted the activation of Phagosome Formation, Cardiac β-adrenergic, tRNA Splicing, and PI3K/AKT pathways. The IPA also predicted the inhibition of Cholesterol Biosynthesis, Adrenomedullin, and Inositol Phosphate Compounds pathways in T2DM-related periodontitis. Validation of DEGs confirmed changes in protein expression of *PTPN2*, *PTPN13*, *DHCR24*, *PIK3R2*, *CALCRL*, *IL1RN*, *IL-6R* and *ITGA4* in diseased tissues in diabetic subjects. Thus, these preliminary findings indicate that there are specific genes and functional pathways that may be involved in the pathogenesis of T2DM-related periodontitis.

## Introduction

Diabetes mellitus (DM) has long been documented as a risk factor for periodontitis, a highly prevalent infectious-inflammatory disease that can cause periodontal tissue breakdown and,

**Funding:** PMD received a University of Florida College of Dentistry Faculty grant. The funders had no role in study design, data collection and analysis, decision to publish, or preparation of the manuscript.

**Competing interests:** The authors have declared that no competing interests exist.

ultimately, tooth loss [1, 2]. DM is considered a disease grade modifier by the classification system of periodontal diseases, used as a critical indicator of periodontitis progression [3].

Previous studies have attempted to ascertain the changes in specific transcripts and proteins associated with the pathogenesis of DM-related periodontitis [4]. Overall evidence indicates that chronic hyperglycemia may stimulate the pathophysiological determinants of periodontitis onset and progression, including microvascular dysfunction, oxidative stress, hyperinflammation, and dysregulation of cell dynamics [5]. Nevertheless, the specific genes and biological pathways underlying the pathogenesis of type 2 DM (T2DM)-related periodontitis are still undetermined. To date, most of the studies focusing on the pathogenesis of the T2DM-related periodontitis had restricted gene or protein coverage due to the use of low- to mid-plex or microarray-based techniques, thus, failing to assess the biological processes involved in T2DM-related periodontitis comprehensively.

The emergence of high-throughput technologies and bioinformatic approaches enhanced the possibility of understanding the complexity of the eukaryotic transcriptome related to several diseases and identifying broader gene expression profiles, specific signaling pathways, and novel transcript variants [6]. To the best of our knowledge, there are no previous studies on whole transcriptome datasets from human gingival biopsies of patients with periodontitis and T2DM. Thus, the aim of this study is to assess the changes in the whole transcriptome that occur from periodontal health to periodontitis in patients with T2DM. We hypothesize that diabetic and non-diabetic patients exhibit significant differences in gene expression profiles that may translate into different immunoinflammatory responses and, ultimately, distinct susceptibility to periodontal breakdown. The transcriptional characterization of the T2DM-related periodontitis is important to identify key genes and signaling pathways that can be used as targets for diagnosis, preventive, and therapeutic strategies in the future.

## Material and methods

### Population

In this cross-sectional study, 36 patients (45–85 years old) were selected from a pool of patients seeking treatment at the University of Florida's Dental Clinics from January 2020 to January 2022. The study was approved by the Institutional Review Board (IRB201901176), and written informed consent was obtained from all eligible individuals. All experiments were performed in accordance with the Declaration of Helsinki (Ethical Principles for Medical Research Involving Human Subjects) and relevant guidelines and regulations. The Strengthening the Reporting of Observational Studies in Epidemiology (STROBE) guidelines were adopted for study conducting and reporting findings.

Based on periodontal and DM status, patients were assigned into one of the following four groups: non-diabetic patients without periodontitis (HH; n = 9), non-diabetic patients with periodontitis (HP; n = 9), T2DM patients without periodontitis (DH; n = 9), and T2DM patients with periodontitis (DP: n = 9). The periodontally-healthy patients had no history of periodontitis, <10% of sites with marginal bleeding (MB) or bleeding on probing (BoP), and probing depths (PD) ≤3mm (assuming no pseudopockets) [7]. Furthermore, the periodontally-healthy patients should have needed at least one crown lengthening surgery or tooth extraction due to non-periodontitis reasons. The periodontitis patients had localized or generalized stage III or IV grade B or C periodontitis [8] and a referral for at least one tooth extraction due to severe untreated periodontitis. The diabetic patients were required to have a diagnosis of T2DM confirmed by a physician, dating from at least the last three years, and confirmed by HbA1c levels ≥7% and a fasting plasma glucose (FPG) >99 mg/dl in last two

months. The non-diabetic patients were required to have no history of pre-DM or DM, confirmed by an HbA1c <5.5% and an FPG <99 mg/dl.

The exclusion criteria were history of smoking (current or past); pregnancy; lactation; use of antibiotics, anti-inflammatory, immunosuppressant, or other immune regulators within six months; use of bone metabolism-related medications (e.g., bisphosphonates, or denosumab); subgengival periodontal treatment in the past 12 months; regular use of mouth rinses containing antimicrobials; and other systemic conditions that could affect periodontitis progression [9].

The demographic and clinical data of the participants were obtained from their records, including age, sex, HbA1C levels, duration of DM, and full-mouth periodontal parameters. In order to have accuracy in the periodontal diagnosis of the sampled teeth, periodontal parameters were recorded around the sampled teeth by the same trained and calibrated examiner (P. M.D). The study examiner participated in a previous calibration exercise, and the standard error of measurement was estimated. Intra-examiner variability was recorded as 0.20 mm for PD and 0.23 mm for CAL.

## Tissue collection

Gingival biopsies (surgical waste specimens), including oral, junctional, and sulcular epitheliums, and connective tissue were obtained during surgical procedures. In patients with periodontitis, the biopsies were obtained from sites with a PD and clinical attachment level (CAL) ≥5 mm and with BoP, mobility, and/or bone loss of more than half of the root during tooth extraction due to severe periodontitis. In the periodontally-healthy patients, the biopsies were obtained from teeth with a PD ≤3 mm, no bone loss, and no BoP or marginal bleeding during crown lengthening surgery or tooth extraction due to non-periodontitis reasons. Briefly, internal bevel and intrasulcular incisions were performed around selected teeth using a scalpel blade. Subsequently, the collar of gingival tissue was carefully removed. After removing the blood from the biopsies by washing them in a saline solution, samples were immediately stored in an RNA later solution (Ambion Inc., Austin, TX, USA) at −80oC for future RNA extraction.

## RNA extraction

Total RNA was isolated from the biopsies using the Trizol method (TRIzol™ Reagent, Thermo Fisher Scientific, Waltham, MA, USA), according to the manufacturer's instructions. RNA samples were resuspended in diethylpyrocarbonate-treated (DEPC) water and stored at −80oC. The RNA concentration was determined by the optical density using a micro-volume spectrophotometer (Nanodrop 1000, Nanodrop Technologies LLC, Wilmington, NC, USA). The total RNA was DNase treated (Turbo DNA-free, Ambion Inc., Austin, TX, USA) to rid the RNA samples of contaminating DNA.

## Library construction, sequencing, and bioinformatics

**Illumina sequencing library construction.** The RNA-seq library construction was performed at the University of Florida's ICBR Gene Expression and Genotyping Core Facility (RRID: SCR_019145). The RNA samples were measured by the QUBIT fluorescent method (Invitrogen, Waltham, MA, USA) and Agilent Bioanalyzer (Agilent, Santa Clara, CA, USA). An amount of 500 ng of protein-free, total RNA was used for the library construction using the reagents provided in the NEBNext rRNA depletion kit (catalog # E6350) and the Ultra Directional RNA Library Prep Kit for Illumina (New England Biolabs, Ipswich, MA, USA), following the manufacturer's recommendations. Two microliters of probe hybridization buffer were added to the total RNA samples, followed by RNase H and DNAase I digestion. Subsequently, the RNA was purified using NEBNext RNA purification beads (New England Biolabs,

catalog # E6350). This step was followed by the RNA library construction using the NEBNext Ultra II Directional Lib Prep (New England Biolabs, catalog # E7760), according to the manufacturer's user guide. Briefly, RNA was fragmented in a solution containing divalent cations and incubated at 94oC, followed by first-strand cDNA synthesis using reverse transcriptase and oligo dT primers. The synthesis of the ds-cDNA was done using the 2nd strand master mix provided in the kit, followed by end-repair and dA-tailing. Next, Illumina adaptors were ligated to the samples. Finally, the library was amplified, followed by purification with AMPure beads (Beckman Coulter, Pasadena, CA, USA, catalog # A63881). The library size and mass were assessed by analysis in an Agilent TapeStation using a DNA5000 Screen Tape (Agilent, Santa Clara, CA, USA). A 200–1000 broad library peak was observed, with the highest peak at around 500 bp. Quantitative PCR was used to validate the library's functionality, using the KAPA library quantification kit (Kapa Biosystems, Wilmington, MA, USA catalog # KK4824) and monitored on a BioRad CFX 96 real-time PCR system (Bio-Rad Laboratories, Hercules, CA, USA). Eighteen barcoded samples were pooled equimolarly for simultaneous sequencing, as described below.

**Illumina NovaSeq6000 sequencing.** The sequencing run was performed at the University of Florida's ICBR NextGen DNA Sequencing Core Facility (RRID: SCR_019152). Briefly, normalized libraries were submitted to the Illumina Free Adapter Blocking (FAB) Reagent protocol (Illumina, San Diego, CA, USA, catalog # 20024145) to minimize the presence of adaptor-dimers and index hopping rates. The library pool was diluted to 0.8 nM and sequenced on the one S4 flow cell lane (2×150 cycles) of an Illumina NovaSeq6000 (Illumina, San Diego, CA, USA). The instrument's computer utilized NovaSeq Control Software v1.6. Cluster and SBS consumables were v1.5. The final loading concentration of the library was 120 pM with a 1% PhiX spike-in control. One lane generated approximately 2.5 billion paired-end reads (∼750 Gb) with an average Q30% greater or equal to 92.5% and a Cluster PF of 85.4%. FastQ files were generated using the BCL2fastQ function in the Illumina BaseSpace portal. An average of 75 million demultiplexed, paired-end reads were used for the data analyses.

**Sequence quality control.** The quality of the RNA-Seq sequence data was first evaluated using FastQC (FASQC) [10] prior to further downstream analysis. Low-quality sequences were trimmed, and poor-quality reads were removed using the Trimmomatic tool [11].

**Differential gene expression analysis.** Data analyses were performed at the University of Florida's ICBR Bioinformatics core. A Star Aligner [12] was used to map the high-quality single-end reads to the genome, GRCh38 (https://useast.ensembl.org/Homo_sapiens/Info/Index/). The gene expression was obtained using RSEM [13]. The expected read counts and fragments per kilobase of transcript per million mapped (FPKM) were extracted for further analysis. The estimated read counts were used as the input for edgeR [14] to perform the differential gene expression analysis. The exact test was developed to identify DEGs, and the thresholds were set at an FDR of 0.05 and a fold change of greater than or equal to 2. Pairwise comparisons were made between the healthy controls and periodontitis in the non-diabetic and diabetic patients (HP versus HH and DP versus DH). Prior to the DE analysis, a PCA was performed to identify outlier samples. No obvious outlier samples were found.

**Pathway analysis.** An Ingenuity Pathway Analysis (IPA) (Qiagen, Redwood City, CA) was performed on the DEGs expressed exclusively in the diseased tissues in diabetic and non-diabetic patients to identify the canonical pathways dysregulated in periodontitis in the presence and absence of T2DM. The transcripts that were identified to be differently expressed in the tissues with periodontitis compared with the tissues with periodontal heath in non-diabetic (HP versus HH) and diabetic patients (DP versus DH), along with their respective fold changes, were input into the IPA, for subsequent bioinformatics analysis. The Canonical Pathways were used for interpreting the functions of the DEGs. For the Ingenuity Canonical

Pathway analysis, a −log (P-value) greater than 2 was taken as the significant threshold for the diabetic and non-diabetic groups. A Z-score over 2 was defined as the threshold of significant activation, while a Z-score under −2 was defined as the threshold of significant inhibition for patients with T2DM. No significant signaling pathways in the non-diabetic patients presented a Z-score over 2 or under −2. Therefore, a Z-score under −1.3 was defined as the threshold of significant inhibition for the non-diabetic patients.

**Clinical and demographic data analyses.** The mean percentages of sites with plaque and BoP, the mean full-mouth PD and CAL, the mean PD and CAL of the sampled teeth, the HbA1c levels, and the duration of DM were computed for each subject and, subsequently, across the groups. Data were examined for normality by the Shapiro-Wilk test, and parametric methods were used as data achieved a normal distribution. The significance of differences for age, HbA1c, and clinical parameters was compared by a one-way ANOVA, followed by a Tukey test. A t-test was used to compare the duration of DM between both diabetic groups. The significance of differences for sex was compared by a Chi-square test. The level of significance for all these analyses was set at 5%.

**IHC.** IHC experiments on four tissues per group were performed in an independent set of patients to confirm the transcriptome findings at the protein level. The selection of the genes for validation was performed based on their biological relevance within the representative signaling pathways observed in the diseased tissues of patients with T2DM and the magnitude of difference in mRNA levels between healthy and disease tissues in diabetic and non-diabetic patients. Gingival samples were collected as described above and fixed in 10% neutral-buffered formalin. Four micrometers-thick formalin-fixed paraffin-embedded (FFPE) sections were set on positively charged glass slides. All the staining steps were performed on a Leica RX Autostainer (Leica Biosystems, Wetzlar, Germany), following the vendor's instructions. The slides were de-paraffinized, re-hydrated, and heat-induced antigen retrieval treated with either epitope retrieval (ER)1 or ER2 buffers for 20 minutes. The slides were treated with hydrogen peroxide to eliminate endogenous peroxidases for five minutes. Afterward, the sections were incubated with primary antibodies against PTPN2 (1:500, polyclonal, rabbit, PA582516, ThermoFisher Scientific, Waltham, MA, USA), PTPN13 (1:500, polyclonal, rabbit, 25944-1-AP, Proteintech, Rosemont, IL, USA), DHCR24 (1:750, polyclonal, rabbit, 10471-1-AP), PIK3R2 (1:100, monoclonal, mouse, 67644-1-Ig, Proteintech, Rosemont, IL, USA), CALCRL (1:400, polyclonal, rabbit, PA550644, ThermoFisher Scientific, Waltham, MA, USA), IL1RN (1:2000, monoclonal, mouse, CF803396, ThermoFisher Scientific, Waltham, MA, USA), IL-6R alpha (1:350, polyclonal, rabbit, 23457-1-AP, Proteintech, Rosemont, IL, USA), and ITGA4 (1:500, polyclonal, rabbit, 10471-1-AP, ThermoFisher Scientific, Waltham, MA, USA) for 30 minutes. The sections were then treated with an HRP Polymer (Leica Biosystems, Wetzlar, Germany) for eight minutes. The specific reaction for each antibody was observed through the application of 3,3'diaminobenzidine (DAB) for ten minutes. The sections were counter-stained with hematoxylin (Leica Biosystems, Wetzlar, Germany), dehydrated through graded ethanol, cleared in xylene, and mounted on slides with the aid of Cytoseal mounting media (Richard-Allan Scientific, San Diego, CA, USA). The negative controls were obtained by the omission of the primary antibodies. Results are presented in a descriptive form, including the description of the histological picture regarding which cells or tissue components were immunopositive, patterns of staining (membrane, cytoplasm, or nuclear), and intensity of IHC expression (weak, moderate, or strong).

## Results

The mean age and frequency of sex did not differ among the groups ($P > 0.05$; S1 Table). The periodontal parameters were significantly higher in both groups with periodontitis compared with the groups without periodontitis ($P < 0.05$; S1 Table). There were no differences between non-diabetic patients without periodontitis (HH) and T2DM patients without periodontitis (DH) and between non-diabetic patients with periodontitis (HP) and T2DM patients with periodontitis (DP) in relation to the periodontal parameters ($P > 0.05$; S1 Table)

### Differentially expressed genes (DEGs) analysis

S1 Fig presents the Venn diagram of the differentially expressed genes (DEGs) in the non-diabetic (HH vs. HP) and diabetic (DH vs. DP) patients exclusively and the overlapping DEGs between the two groups. Four hundred and twenty DEGs (S2 Table; 211 upregulated and 209 downregulated) were identified in tissues with periodontitis compared with those with periodontal health in non-diabetic patients from the 16,164 background genes. One thousand five hundred and sixty-three DEGs (S3 Table; 654 upregulated and 909 downregulated) were identified in tissues with periodontitis compared with those with periodontal health in diabetic patients from the 16,299 background genes. One thousand two hundred and forty-two DEGs were concomitantly expressed in tissues with periodontitis in the non-diabetic and diabetic patients (S4 Table; 708 upregulated and 534 downregulated).

The top 15 up and downregulated DEGs in the non-diabetic (HP vs. HH) and diabetic (DP vs. DH) groups are listed in Table 1. Overall, immunoglobulin (Ig)-related genes were highly expressed in the non-diabetic and diabetic patients. Excluding the Ig-related DEGs, inhibin βE (*INHBE*) and C-X-C motif chemokine ligand 5 (*CXCL5*) were the most upregulated DEGs in non-diabetic and diabetic patients, respectively. Adhesion G protein-coupled receptor G7 (*ADGRG7*) and *AL356488.2* were the DEGs with the strongest downregulation in non-diabetic and diabetic patients, respectively. Five genes from the late cornified envelope (LCE) family were amongst the top 15 downregulated DEGs in tissues with periodontitis in diabetic patients.

**Ingenuity Pathway Analysis (IPA).** The list of ingenuity canonical pathways related to the DEGs that were uniquely expressed in non-diabetic patients is presented in S5 Table. Twenty-one ingenuity canonical pathways were identified in non-diabetics by applying a −log (*P*-value) threshold of greater than two. The top ten representative pathways ranked according to the −log (*P*-value) are presented in Fig 1A. B Cell Receptor Signaling was the highest-ranking signaling pathway in the non-diabetic patients. None of the significant signaling pathways in non-diabetic patients presented a Z-score over 2 or under −2. Considering a Z-score of less than −1.3 as significant, the Tumor Microenvironment and PI3K Signaling in B Lymphocytes pathways were significantly inhibited (Fig 1B and 1C). Within these enriched pathways, *FOS*, *JUN*, activating transcription factor 3 (*ATF3*), and matrix metalloproteinase (MMP)-13 were the DEGs with a log2 fold change (FC) less than or equal to −1.5.

The list of ingenuity canonical pathways related to the DEGs exclusively expressed in diabetic patients is presented in S6 Table. Forty-three ingenuity canonical pathways were identified by applying a −log (*P*-value) threshold of greater than two. The top ten signaling pathways are presented in Fig 2A. Granulocyte Adhesion and Diapedesis was the highest-ranking pathway (−log [*P*-value] = 5.55), followed by Phagosome Formation (−log [*P*-value] = 4.75) and Atherosclerosis Signaling (−log [*P*-value] = 4.55). Taking Z-scores greater than 2 and less than −2, Phagosome Formation, Cardiac β-adrenergic, tRNA, and PI3K/AKT signaling pathways were significantly activated. In contrast, the Superpathway of Cholesterol Biosynthesis,

**Table 1. Top 15 downregulated and upregulated DEGs in periodontitis in relation to periodontal health in non-diabetic and in patients with T2DM exclusively.**

| | Downregulated | | | | Upregulated | | | |
|---|---|---|---|---|---|---|---|---|
| | Gene | Q value | P value | Log2 FC | Gene | Q value | P value | Log2 FC |
| DEGs exclusively in non-diabetics (HP vs HH) | ADGRG7 | 6.08E-05 | 1.47E-06 | -9.65 | IGKV1D-17 | 0.0071315 | 0.000790428 | 4.80 |
| | AADAC | 0.00024201 | 9.12E-06 | -7.02 | IGLV10-54 | 0.002906844 | 0.000228862 | 4.67 |
| | TYRP1 | 8.23E-06 | 1.17E-07 | -5.22 | IGKV5-2 | 9.14E-06 | 1.34E-07 | 4.39 |
| | CCER2 | 0.0001552 | 4.92E-06 | -4.77 | MTND2P28 (lnc) ENSG00000225630 | 0.001346243 | 8.27E-05 | 3.75 |
| | AC036176.3 (lnc) ENSG00000269989 | 3.60E-05 | 7.64E-07 | -4.10 | IGKV6D-21 | 1.07E-07 | 4.19E-10 | 3.54 |
| | DCT | 0.00022835 | 8.50E-06 | -3.76 | IGHV4-31 | 0.000355956 | 1.51E-05 | 3.24 |
| | PAX1 | 0.00143649 | 8.98E-05 | -3.32 | INHBE | 0.000189732 | 6.60E-06 | 3.20 |
| | AC073896.1 | 0.00016101 | 5.27E-06 | -3.20 | AC026202.2 (lnc) ENSG00000233912 | 1.74E-06 | 1.65E-08 | 2.99 |
| | TM4SF19 | 0.00018363 | 6.28E-06 | -3.20 | CBSL | 0.009032748 | 0.001093897 | 2.97 |
| | FOSB | 1.74E-06 | 1.64E-08 | -2.93 | IGLV3-9 | 0.003098993 | 0.000247676 | 2.89 |
| | ARC | 0.00528802 | 0.00051847 | -2.92 | CU633967.1 (lnc) ENSG00000274333 | 2.61E-07 | 1.48E-09 | 2.79 |
| | NPIPA8 | 0.03595036 | 0.00706707 | -2.86 | AMPD1 | 1.23E-05 | 1.94E-07 | 2.77 |
| | SMARCA5-AS1 (lnc) ENSG00000245112 | 0.00016881 | 5.61E-06 | -2.82 | IGKV2D-30 | 0.00849601 | 0.00100104 | 2.73 |
| | AC126773.2 (lnc) ENSG00000260577 | 0.0007186 | 3.67E-05 | -2.77 | MAP3K19 | 3.52E-05 | 7.38E-07 | 2.70 |
| | AC106886.2 (lnc) ENSG00000260899 | 0.0001565 | 4.99E-06 | -2.71 | JCHAIN | 8.20E-05 | 2.19E-06 | 2.60 |
| DEGs exclusivaly in diabetics (DP vs DH) | SCARNA2 (lnc) ENSG00000270066 | 5.91E-26 | 2.18E-29 | -10.8 | IGHV3-20 | 4.21E-09 | 8.12E-11 | 7.12 |
| | LCE2C | 7.91E-08 | 2.34E-09 | -9.89 | IGHV1-69 | 0.00040398 | 4.51E-05 | 5.46 |
| | AC105001.2 | 6.80E-13 | 4.26E-15 | -9.88 | IGHV3-72 | 1.36E-08 | 3.08E-10 | 5.44 |
| | LCE2A | 9.99E-06 | 5.82E-07 | -9.05 | IGHV3-33 | 5.08E-10 | 7.47E-12 | 4.80 |
| | LCE2B | 2.43E-06 | 1.15E-07 | -8.93 | IGKV1D-16 | 0.0001974 | 1.94E-05 | 4.66 |
| | AC138811.2 | 1.47E-07 | 4.67E-09 | -8.73 | IGHV3-48 | 3.36E-13 | 1.94E-15 | 4.22 |
| | LCE2D | 3.71E-05 | 2.71E-06 | -8.688 | IGHV3-74 | 4.01E-16 | 8.12E-19 | 4.04 |
| | SMAD1-AS1 (lnc) ENSG00000250902 | 0.00034329 | 3.72E-05 | -8.63 | IGKV2D-29 | 1.29E-05 | 7.76E-07 | 3.48 |
| | SENP3-EIF4A1 (lnc) ENSG00000277957 | 1.47E-05 | 9.02E-07 | -8.30 | AC244226.1 | 0.00201771 | 0.00029938 | 3.46 |
| | ATP12A | 5.27E-06 | 2.79E-07 | -8.25 | CXCL5 | 0.0001365 | 1.25E-05 | 3.45 |
| | KLRG2 | 1.98E-05 | 1.29E-06 | -7.96 | IGHV3-66 | 0.028498749 | 0.007819436 | 3.21 |
| | AP000350.6 (lnc) ENSG00000273295 | 6.86E-06 | 3.79E-07 | -6.57 | IGHV1-46 | 2.04E-05 | 1.33E-06 | 3.18 |
| | LCE3C | 2.03E-06 | 9.40E-08 | -5.79 | IGLV3-21 | 0.003652181 | 0.000613812 | 3.18 |
| | KRT76 | 0.00013426 | 1.23E-05 | -5.19 | P2RX5 | 7.01E-16 | 1.55E-18 | 3.15 |
| | PLA2G2F | 3.51E-05 | 2.54E-06 | -4.97 | GDF15 | 5.80E-07 | 2.25E-08 | 3.02 |

Adrenomedullin (ADM), and Superpathway of Inositol Phosphate Compounds (SIPC) were inhibited in tissues with periodontitis in the diabetic patients (Fig 2B and 2C).

Table 2 presents the DEGs that function within the abovementioned canonical pathways in diabetic patients. The upregulated DEGs that function in the Phagosome Formation pathway included mostly adhesion G protein-coupled receptors (GPCRs), complement-related factors, chemokine, pathogen pattern-recognition and Ig receptors, and integrin (ITG) subunits. The downregulated DEGs involved in this pathway were comprised mainly of GPCRs, mitogen-

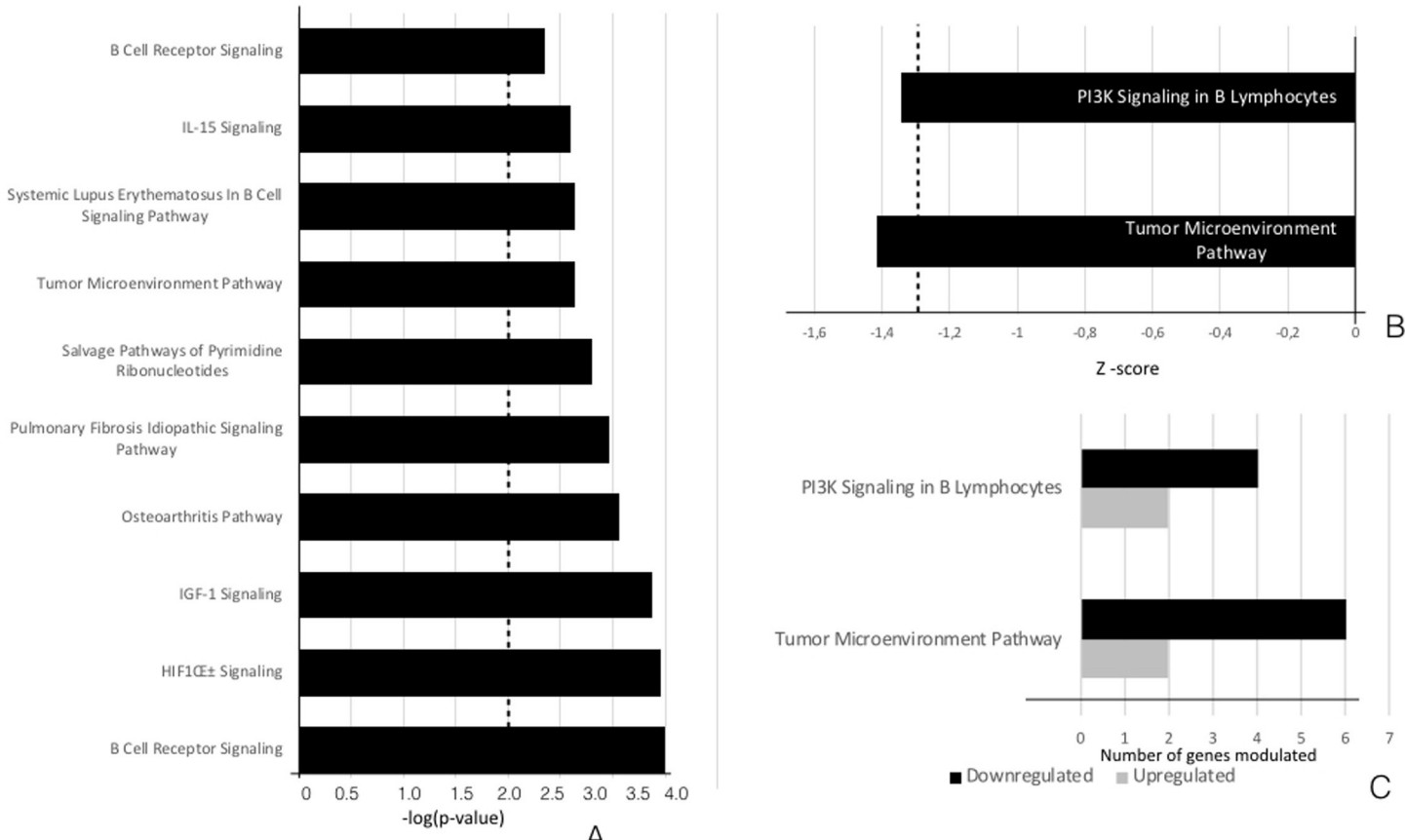

**Fig 1. IPA analysis of the canonical pathways related to the DEGs altered in tissues with periodontitis compared with tissues with periodontal health in non-diabetic patients.** (A) The top ten dysregulated canonical signaling pathways are presented in descending order of statistical significance of −log(*P*-values). (B) The canonical pathways that were predicted to be modulated in the non-diabetic patients, as defined by a −log(*P*-value) >2 and Z-score <−1.3 (inhibited). (C) The number of upregulated and downregulated DEGs involved in each of the representative canonical pathways.

activated protein kinase (MAPK), and phospholipases (PL). ITG subunits were the predominant upregulated gene group, while interleukin (IL) receptors and phosphatase-related components were the leading downregulated DEGs functioning in PI3K/AKT signaling. Most of the upregulated genes in the Cardiac β-Adrenergic and tRNA Splicing pathways were phosphodiesterases. Of the 12 genes downregulated in Cardiac β-Adrenergic Signaling, six were phosphatase-related, and two were calcium voltage-gated channel auxiliary-related genes. The primary downregulated genes observed in the ADM pathway were comprised of MAPKs, Phospholipase C (PLC), and transcription factors. Notably, in the ADM pathway, *IL1A* was upregulated while IL1 receptor antagonist (*IL1RN*) was downregulated in the diabetic patients. The mRNA for different enzyme functional classes were downregulated in Superpathway of Cholesterol Biosynthesis signaling. Phosphatase and PL genes were generally downregulated in SIPC signaling.

## Validation of transcriptome data by immunohistochemistry (IHC)

The immunohistochemistry (IHC) are shown in Figs 3 and 4.

Protein tyrosine phosphatase non-receptor (PTPN) 2 showed slightly decreased expression in the DH group compared with the HH group. PTPN2 expression was increased in both

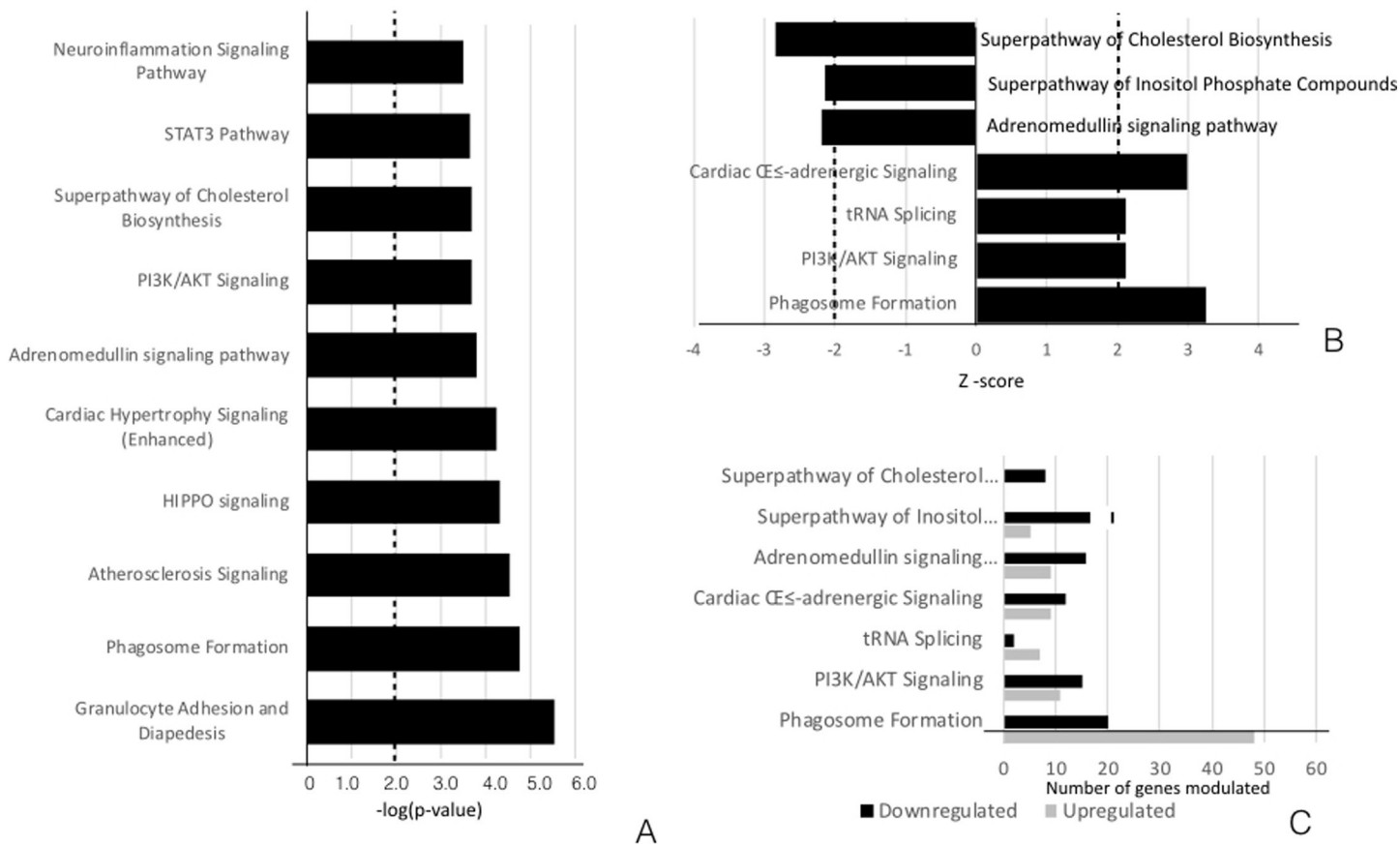

**Fig 2. IPA analysis of the canonical pathways related to the DEGs altered in tissues with periodontitis compared with tissues with periodontal health in patients with T2DM.** (A) The top ten dysregulated canonical signaling pathways are presented in descending order of statistical significance of −log($P$-values). (B) The canonical pathways that were predicted to be modulated in the non-diabetic patients, as defined by a −log($P$-value) >2 and Z-score >2 (activated) or <−2 (inhibited). (C) The number of upregulated and downregulated DEGs involved in each of the representative canonical pathways.

periodontitis groups but was more pronounced in the DP group, particularly in the connective tissue (Fig 3). PTPN13 had a noticeable nuclear and cytoplasmatic protein expression reduction in the DP group (Fig 3). DHCR24 presented with considerable expression in the cytoplasm and nucleus in the basal and supra-basal layers of epithelial and endothelial cells in the HH group. However, its expression was significantly reduced in the DH group, especially in the basal cell layer. The DP group showed the lowest expression levels of the DHCR24 protein in intensity and distribution (Fig 3). The DH group presented noticeably lower Phosphoinositide-3-Kinase Regulatory Subunit 2 (PIK3R2) protein expression than the HH group. While the PIK3R2 levels decreased in the HP group (mainly in the basal cell layer), the DP group showed increased cytoplasmatic and nuclear PIK3R2 expression in the epithelium and connective tissues (Fig 3).

The calcitonin receptor-like receptor (CALCRL) protein expression was overall stronger in diseased tissues than in healthy tissues (particularly in epithelial and endothelial cells). However, it was even stronger in the DP group, with nuclear and cytoplasmatic distribution. Notably, while CALCRL expression was clearly noticed in the basal cell layers in the HH group, it was barely expressed in the basal layers in the DH group. IL1RN showed intense staining in the cytoplasm and nucleus of all the epithelial cell layers in both periodontally-healthy groups;

**Table 2. The DEGs that are a part of the most critical canonical pathways in patients with T2DM and periodontitis compared with the patients with T2DM and without periodontitis.**

| Upregulated | | | | Downregulated | | | |
|---|---|---|---|---|---|---|---|
| Gene symbol | Gene name | Q value | Log2 FC | Gene symbol | Gene name | Q value | Log2 FC |
| Phagosome Formation: -log (p-value) = 4.75; z-score = 3.25 | | | | | | | |
| ABHD3 | Abhydrolase Domain Containing 3, Phospholipase | 8.31E-05 | 1.23 | ADGRF4 | Adhesion G Protein-Coupled Receptor F4 | 0.001502735 | -1.34 |
| ADGRD1 | Adhesion G Protein-Coupled Receptor D1 | 0.000221199 | 1.49 | ADGRG1 | Adhesion G Protein-Coupled Receptor G1 | 7.43E-05 | -1.14 |
| ADGRE5 | Adhesion G Protein-Coupled Receptor E5 | 0.000103337 | 1.01 | AP1M2 | Adaptor Related Protein Complex 1 Subunit Mu 2 | 0.01001205 | -1.10 |
| ADGRF5 | Adhesion G Protein-Coupled Receptor F5 | 2.95E-10 | 1.47 | CELSR1 | Cadherin EGF LAG Seven-Pass G-Type Receptor 1 | 2.25E-05 | -1.23 |
| ADGRL4 | Adhesion G Protein-Coupled Receptor L4 | 7.13E-08 | 1.69 | CCR7 | C-C Motif Chemokine Receptor 7 | 0.020276507 | -1.20 |
| AP1S2 | Adaptor Related Protein Complex 1 Subunit Sigma 2 | 1.46E-06 | 1.35 | F2RL1 | F2R Like Trypsin Receptor 1 | 0.000861457 | -2.03 |
| APBB1IP | Amyloid Beta Precursor Protein Binding Family B Member 1 Interacting Protein | 2.58E-05 | 1.41 | LGR4: | Leucine Rich Repeat Containing G Protein-Coupled Receptor 4 | 3.95E-05 | -1.08 |
| APLNR | Apelin Receptor | 1.22E-05 | 1.43 | MAPK15 | Mitogen-Activated Protein Kinase 15 | 0.010420729 | -1.52 |
| AVPR1A | Arginine Vasopressin Receptor 1A | 0.000286265 | 1.55 | MAPK7 | Mitogen-Activated Protein Kinase 7 | 6.82E-05 | -1.17 |
| C3 | Complement C3 | 0.000197404 | 1.44 | MCHR1 | Melanin Concentrating Hormone Receptor 1 | 0.036117633 | -1.69 |
| C5AR1 | Complement C5a Receptor 1 | 0.000391725 | 1.57 | P2RY11 | Purinergic Receptor P2Y11 | 0.015989797 | -1.06 |
| CALCRL | Calcitonin receptor-like receptor | 9.96E-06 | 1.48 | PIK3R2 | Phosphoinositide-3-Kinase Regulatory Subunit 2 | 3.02E-07 | -1.24 |
| CCR1 | C-C Motif Chemokine Receptor 1 | 0.006780284 | 1.15 | P2RY2 | Purinergic Receptor P2Y2 | 4.27E-07 | -1.71 |
| CCR2 | C-C Motif Chemokine Receptor 2 | 0.000328093 | 1.40 | PLA2G2F | Phospholipase A2 Group IIF | 3.51E-05 | -4.97 |
| CLEC4E | C-Type Lectin Domain Family 4 Member E | 0.005497415 | 1.83 | PLA2G4B | Phospholipase A2 Group IVB | 0.001044935 | -1.68 |
| CR1 | Complement receptor 1 | 8.19E-05 | 1.73 | PLD2 | Phospholipase D2 | 0.000178173 | -1.11 |
| CR2 | Complement C3d Receptor 2 | 0.035193615 | 1.61 | SMO | Smoothened, Frizzled Class Receptor | 2.48E-05 | -1.38 |
| EDNRB | Endothelin receptor type B | 0.002427503 | 1.06 | SRC | SRC Proto-Oncogene, Non-Receptor Tyrosine Kinase | 3.84E-05 | -1.08 |
| F2RL2 | Coagulation Factor II Thrombin Receptor Like 2 | 3.37E-05 | 1.47 | VAV3 | Vav Guanine Nucleotide Exchange Factor 3 | 0.000301819 | -1.16 |
| FCER1G | Fc Fragment of IgE Receptor Ig | 0.02219775 | 1.01 | VIPR1 | Vasoactive Intestinal Peptide Receptor 1 | 0.01962491 | -1.15 |
| FCGR1A | Fc Fragment of IgG Receptor Ia | 0.004828328 | 1.27 | | | | |
| FCGR1B | Fc Fragment of IgG Receptor Ib | 0.008226445 | 1.73 | | | | |
| FCGR2C | Fc Fragment of IgG Receptor IIc (Gene/ Pseudogene) | 0.010786066 | 1.33 | | | | |
| FZD3 | Frizzled Class Receptor 3 | 1.49E-07 | 1.38 | | | | |
| GPR162 | G Protein-Coupled Receptor 162 | 0.023897665 | 1.15 | | | | |
| GPR182 | G Protein-Coupled Receptor 182 | 0.00076005 | 1.70 | | | | |
| GPR180 | G Protein-Coupled Receptor 180 | 0.00169883 | 1.06 | | | | |
| GPR22 | G Protein-Coupled Receptor 22 | 0.008043412 | 1.10 | | | | |
| GPR34 | G Protein-Coupled Receptor 34 | 0.001864425 | 1.29 | | | | |
| GPR4 | G Protein-Coupled Receptor 4 | 0.02343798 | 1.00 | | | | |
| GPR52 | G Protein-Coupled Receptor 52 | 0.008604626 | 1.24 | | | | |
| GPR65 | G Protein-Coupled Receptor 65 | 0.001368858 | 1.60 | | | | |
| HRH2 | Histamine Receptor H2 | 0.006393526 | 1.35 | | | | |
| ITGA1 | Integrin Subunit Alpha 1 | 3.91E-05 | 1.21 | | | | |
| ITGA4 | Integrin Subunit Alpha 4 | 0.003952353 | 1.05 | | | | |

*(Continued)*

**Table 2.** (*Continued*)

| Upregulated | | | | Downregulated | | | |
|---|---|---|---|---|---|---|---|
| Gene symbol | Gene name | Q value | Log2 FC | Gene symbol | Gene name | Q value | Log2 FC |
| *ITGAM* | Integrin Subunit Alpha M | 0.002551803 | 1.29 | | | | |
| *ITGAX* | Integrin Subunit Alpha X | 0.007791008 | 1.07 | | | | |
| *LBP* | Lipopolysaccharide Binding Protein | 0.011434944 | 2.43 | | | | |
| *LYN* | LYN Proto-Oncogene, Src Family Tyrosine Kinase | 0.000863943 | 1.01 | | | | |
| *MSR1* | Macrophage Scavenger Receptor 1 | 0.001405248 | 1.24 | | | | |
| *PLA2G4C* | Phospholipase A2 Group IVC | 0.004463955 | 1.23 | | | | |
| *PTGFR* | Prostaglandin F Receptor | 0.00014384 | 1.41 | | | | |
| *S1PR3* | Sphingosine-1-Phosphate Receptor 3 | 0.002548839 | 1.33 | | | | |
| *S1PR4* | Sphingosine-1-Phosphate Receptor 4 | 0.000299743 | 1.45 | | | | |
| *TLR2* | Toll Like Receptor 2 | 0.00426382 | 1.01 | | | | |
| *TLR4* | Toll Like Receptor 4 | 5.63E-07 | 1.66 | | | | |
| *TLR8* | Toll Like Receptor 8 | 0.013887157 | 1.39 | | | | |
| *WIPF1* | WAS/WASL Interacting Protein Family Member 1 | 0.000146684 | 1.13 | | | | |
| **PI3K/AKT Signaling: -log (p-value) = 3.69; z-score = 2.11** | | | | | | | |
| *BCL2A1* | BCL2 Related Protein A1 | 0.001296469 | 1.66 | *CCND1* | Cyclin D1 | 0.000369428 | -1.00 |
| *CDKN1B* | Cyclin Dependent Kinase Inhibitor 1B | 3.28E-07 | 1.22 | *IKBKE* | Inhibitor Of Nuclear Factor Kappa B Kinase Subunit Epsilon | 4.78E-06 | -1.19 |
| *FOXO1* | Forkhead Box O1 | 1.69E-05 | 1.01 | *IL12RB2* | Interleukin 12 Receptor Subunit Beta 2 | 0.000404413 | -1.79 |
| *GDF15* | Growth Differentiation Factor 15 | 5.80E-07 | 3.02 | *IL17RC* | Interleukin 17 Receptor C | 1.80E-05 | -1.36 |
| *IL18RAP* | Interleukin 18 Receptor Accessory Protein | 0.00028751 | 2.10 | *IL17RD* | Interleukin 17 Receptor D | 0.000389797 | -1.15 |
| *IL6R* | Interleukin 6 Receptor | 2.05E-08 | 1.30 | *IL20RA* | Interleukin 20 Receptor Subunit Alpha | 0.015230566 | -1.06 |
| *ITGA1* | Integrin Subunit Alpha 1 | 3.91E-05 | 1.21 | *IL20RB* | Interleukin 20 Receptor Subunit Beta | 0.000300872 | -1.12 |
| *ITGA4* | Integrin Subunit Alpha 4 | 0.003952353 | 1.05 | *IL22RA1* | Interleukin 22 Receptor Subunit Alpha 1 | 5.43E-07 | -2.26 |
| *ITGAM* | Integrin Subunit Alpha M | 0.002551803 | 1.29 | *IL4R* | Interleukin 4 Receptor | 6.36E-06 | -1.14 |
| *ITGAX* | Integrin Subunit Alpha X | 0.007791008 | 1.07 | *IL9R* | Interleukin 9 Receptor | 0.00360178 | -2.34 |
| *PTGS2* | Prostaglandin-Endoperoxide Synthase 2 | 0.005655989 | 1.44 | *INPP5J* | Inositol Polyphosphate-5-Phosphatase J | 0.00134853 | -1.78 |
| | | | | *PIK3R2* | Phosphoinositide-3-Kinase Regulatory Subunit 2 | 3.02E-07 | -1.29 |
| | | | | *PPM1J* | Protein Phosphatase, Mg2+/Mn2 + Dependent 1J | 7.81E-05 | -2.40 |
| | | | | *PPP2R2B* | Protein Phosphatase 2 Regulatory Subunit Bbeta | 0.006000897 | -1.74 |
| | | | | *PPP2R3A* | Protein Phosphatase 2 Regulatory Subunit B"Alpha | 1.06E-05 | -1.02 |
| **Cardiac Œ≤-adrenergic Signaling: -log (p-value) = 2.90; z-score = 3.0** | | | | | | | |
| *AKAP12* | A-Kinase Anchoring Protein 12 | 4.23E-06 | 1.37 | *CACNA2D3* | Calcium Voltage-Gated Channel Auxiliary Subunit Alpha2delta 3 | 3.29E-06 | -1.85 |
| *PDE10A* | Phosphodiesterase 10A | 0.000450127 | 1.18 | *CACNB4* | Calcium Voltage-Gated Channel Auxiliary Subunit Beta 4 | 0.001979528 | -1.39 |
| *PDE1A* | Phosphodiesterase 1A | 2.77E-06 | 2.26 | *GDPD3* | Glycerophosphodiester Phosphodiesterase Domain Containing 3 | 0.005953332 | -1.33 |
| *PDE4A* | Phosphodiesterase 4A | 0.000197404 | 1.05 | *GNA11* | G Protein Subunit Alpha 11 | 9.91E-07 | -1.15 |
| *PDE4B* | Phosphodiesterase 4B | 3.64E-05 | 1.30 | *PDE6A* | Phosphodiesterase 6A | 0.032414648 | -1.44 |
| *PDE4D* | Phosphodiesterase 4D | 0.000401816 | 1.02 | *PPM1J* | Protein Phosphatase, Mg2+/Mn2 + Dependent 1J | 7.81E-05 | -2.40 |

(*Continued*)

**Table 2.** (Continued)

| | Upregulated | | | | Downregulated | | |
|---|---|---|---|---|---|---|---|
| Gene symbol | Gene name | Q value | Log2 FC | Gene symbol | Gene name | Q value | Log2 FC |
| PDE5A | Phosphodiesterase 5A | 2.69E-05 | 1.05 | PPP1R14B | Protein Phosphatase 1 Regulatory Inhibitor Subunit 14B | 1.36E-05 | -1.12 |
| PDE7B | Phosphodiesterase 7B | 0.002560175 | 1.27 | PPP1R14C | Protein Phosphatase 1 Regulatory Inhibitor Subunit 14C | 0.000537905 | -1.43 |
| PRKACB | Protein Kinase CAMP-Activated Catalytic Subunit Beta | 1.02E-06 | 1.50 | PPP1R3C | Protein Phosphatase 1 Regulatory Subunit 3C | 2.72E-05 | -1.48 |
| | | | | PPP2R2B | Protein Phosphatase 2 Regulatory Subunit B beta: | 0.006000897 | -1.74 |
| | | | | PPP2R3A | Protein Phosphatase 2 Regulatory Subunit B"Alpha: | 1.06E-05 | -1.02 |
| | | | | PRKAR1B | Protein Kinase CAMP-Dependent Type I Regulatory Subunit Beta | 0.005500148 | -1.03 |
| colspan tRNA Splicing: -log (p-value) = 2.83; z-score = 2.12 | | | | | | | |
| PDE10A | Phosphodiesterase 10A | 0.000450127 | 1.18 | GDPD3 | Glycerophosphodiester Phosphodiesterase Domain Containing 3 | 0.005953332 | -1.33 |
| PDE1A | Phosphodiesterase 1A | 2.77E-06 | 2.26 | PDE6A | Phosphodiesterase 6A | 0.032414648 | -1.44 |
| PDE4A | Phosphodiesterase 4A | 0.000197404 | 1.05 | | | | |
| PDE4B | Phosphodiesterase 4B | 3.64E-05 | 1.30 | | | | |
| PDE4D | Phosphodiesterase 4D | 0.000401816 | 1.02 | | | | |
| PDE5A | Phosphodiesterase 5A | 2.69E-05 | 1.05 | | | | |
| PDE7B | Phosphodiesterase 7B | 0.002560175 | 1.27 | | | | |
| Adrenomedullin signaling pathway: -log (p-value) = 3.80; z-score = -2.2 | | | | | | | |
| BCL2 | BCL2 Apoptosis Regulator | 3.64E-05 | 1.16 | GNA11 | G Protein Subunit Alpha 11 | 9.91E-07 | -1.15 |
| C3 | Complement C3 | 0.000197404 | 1.44 | GUCY2C | Guanylate Cyclase 2C | 0.006407274 | -1.22 |
| CALCRL | Calcitonin receptor-like receptor | 9.96E-06 | 1.48 | IL1RN | Interleukin 1 Receptor Antagonist | 0.029831597 | -1.01 |
| CASP3 | Caspase 3 | 4.78E-07 | 1.22 | IL36A | Interleukin 36 Alpha | 0.002517721 | -1.52 |
| GPR182 | G Protein-Coupled Receptor 182 | 0.000760052 | 1.70 | MAPK13 | Mitogen-Activated Protein Kinase 13 | 5.38E-07 | -1.38 |
| IL1A | Interleukin 1 Alpha | 9.31E-05 | 1.22 | MAPK15 | Mitogen-Activated Protein Kinase 15 | 0.010420729 | -1.52 |
| PLCL2 | Phospholipase C Like 2 | 2.00E-07 | 1.42 | MAPK7 | Mitogen-Activated Protein Kinase 7 | 6.82E-05 | -1.17 |
| PRKACB | Protein Kinase CAMP-Activated Catalytic Subunit Beta | 1.02E-06 | 1.50 | PIK3R2 | Phosphoinositide-3-Kinase Regulatory Subunit 2 | 3.02E-07 | -1.24 |
| RAMP2 | Receptor Activity Modifying Protein 2 | 4.27E-05 | 1.35 | PLCD1 | Phospholipase C Delta 1 | 3.66E-05 | -1.38 |
| | | | | PLCD3 | Phospholipase C Delta 3 | 0.000111424 | -1.22 |
| | | | | PLCH2 | Phospholipase C Eta 2 | 2.34E-08 | -1.75 |
| | | | | PPARG | Peroxisome Proliferator Activated Receptor Gamma | 0.033030069 | -1.31 |
| | | | | PRKAR1B | Protein Kinase CAMP-Dependent Type I Regulatory Subunit Beta | 0.005500148 | -1.03 |
| | | | | RXRA | Retinoid X receptor alpha | 9.18E-05 | -1.08 |
| | | | | TFAP2A | Transcription Factor AP-2 Alpha | 7.30E-08 | -1.73 |
| | | | | TFAP2C | Transcription factor AP-2 gamma | 8.19E-08 | -1.37 |
| Superpathway of Cholesterol Biosynthesis: -log (p-value) = 3.68; z-score = -2.82 | | | | | | | |
| | | | | ACAT2 | Acetyl-CoA Acetyltransferase 2 | 2.67E-06 | -1.61 |
| | | | | DHCR24 | 24-Dehydrocholesterol Reductase | 0.00019843 | -1.44 |
| | | | | DHCR7 | 7-Dehydrocholesterol Reductase | 0.00073273 | -1.04 |
| | | | | MVD | Mevalonate Diphosphate Decarboxylase | 8.06E-06 | -1.53 |
| | | | | MVK | Mevalonate Kinase | 2.19E-05 | -1.47 |
| | | | | PMVK | Phosphomevalonate kinase | 3.19E-06 | -1.10 |

(Continued)

**Table 2.** (Continued)

| Upregulated | | | | Downregulated | | | |
|---|---|---|---|---|---|---|---|
| Gene symbol | Gene name | Q value | Log2 FC | Gene symbol | Gene name | Q value | Log2 FC |
| | | | | SQLE | Squalene Epoxidase | 2.76E-05 | -1.28 |
| | | | | TM7SF2 | Transmembrane 7 Superfamily Member 2 | 5.64E-06 | -1.48 |
| Superpathway of Inositol Phosphate Compounds: -log (p-value) = 2.73; z-score = -2.13 | | | | | | | |
| ALPL | Alkaline Phosphatase, Biomineralization Associated | 2.25E-08 | 1.49 | DUSP14 | Dual Specificity Phosphatase 14 | 0.004823952 | -1.20 |
| EYA4 | EYA Transcriptional Coactivator And Phosphatase 4 | 0.003920438 | 1.25 | EPHX2 | Epoxide Hydrolase 2 | 0.001017046 | -1.06 |
| NUDT4 | Nudix Hydrolase 4 | 7.46E-07 | 1.12 | INPP5J | Inositol Polyphosphate-5-Phosphatase J | 0.00134853 | -1.78 |
| PPP1R16B | Protein Phosphatase 1 Regulatory Subunit 16B | 0.001087609 | 1.02 | IPPK | Inositol-Pentakisphosphate 2-Kinase | 2.59E-06 | -1.37 |
| PTPN2 | Protein Tyrosine Phosphatase Non-Receptor Type 2 | 8.70E-06 | 1.03 | ITPKC | Inositol-Trisphosphate 3-Kinase C | 0.001218582 | -1.25 |
| | | | | NUDT1 | Nudix Hydrolase 1 | 0.003150706 | -1.09 |
| | | | | PIK3R2 | Phosphoinositide-3-Kinase Regulatory Subunit 2 | 3.02E-07 | -1.24 |
| | | | | PLCD1 | Phospholipase C Delta 1 | 3.66E-05 | -1.38 |
| | | | | PLCD3 | Phospholipase C Delta 3 | 0.000111424 | -1.22 |
| | | | | PLCH2 | Phospholipase C Eta 2 | 2.34E-08 | -1.75 |
| | | | | PPFIA3 | PTPRF interacting protein alpha 3 | 0.000150788 | -1.36 |
| | | | | PPP1R14B | Protein Phosphatase 1 Regulatory Inhibitor Subunit 14B | 1.36E-05 | -1.12 |
| | | | | PPP1R14C | Protein Phosphatase 1 Regulatory Inhibitor Subunit 14C | 0.000537905 | -1.43 |
| | | | | PPP1R3C | Protein Phosphatase 1 Regulatory Subunit 3C | 2.72E-05 | -1.47 |
| | | | | PPP2R2B | Protein Phosphatase 2 Regulatory Subunit B beta | 0.006000897 | -1.74 |
| | | | | PPP2R3A | Protein Phosphatase 2 Regulatory Subunit B"Alpha | 1.06E-05 | -1.02 |
| | | | | PTPN13 | Protein Tyrosine Phosphatase Non-Receptor Type 13 | 0.00191405 | -1.24 |
| | | | | SGPP2 | Sphingosine-1-Phosphate Phosphatase 2 | 0.000186653 | -1.26 |
| | | | | SSH3 | Slingshot Protein Phosphatase 3 | 1.47E-07 | -1.45 |
| | | | | PPM1J | Protein Phosphatase, Mg2+/Mn2+ Dependent 1J | 7.81E-05 | -2.40 |

however, it had an even stronger staining in the DH group (Fig 4). The IL1RN protein expression was considerably reduced in the cytoplasm and was more prominent in the nucleus of epithelial cells in the diseased tissues, especially in the DP group (Fig 4). IL6R expression was overall stronger in the periodontitis groups than the non-periodontitis groups; however, it was even more intense in the DP group, particularly in the basal layer of the epithelium and connective tissue (Fig 4). There was a robust ITGA4 expression in the cytoplasm and the membrane of supra-basal cell layers of the epithelium in the HH, HP, and DH groups. Remarkably, the basal cell layer in the DH group exhibited no or mild ITGA4 protein expression. Moreover, the DP group presented a more noticeable ITGA4 expression even in the basal layers of the epithelium and underlying connective tissue cells (Fig 4).

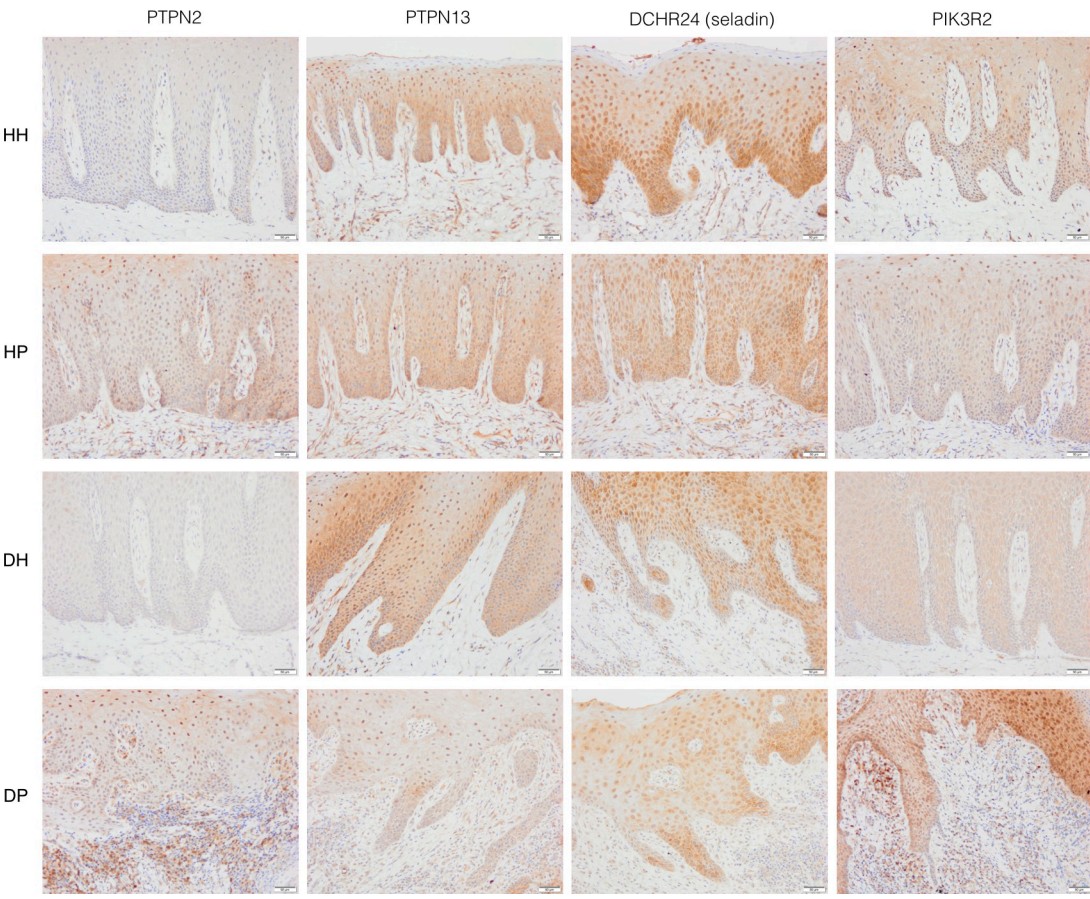

**Fig 3. The immunohistochemical staining using antibodies against PTPN2, PTPN13, DHCR24 (Seladin), and PIK3R2 in representative gingival biopsies from the HH, HP, DH, and DP groups.** Bar length = 50 μm, original magnification × 20. HH: non-diabetic patients without periodontitis; HP: non-diabetic patients with periodontitis; DH: T2DM patients without periodontitis; DP: T2DM patients with periodontitis.

## Discussion

We show here that periodontitis in non-diabetic and diabetic patients is a distinct process at the whole transcript level. T2DM significantly alters the expression of specific genes and signaling pathways involved in immune and inflammatory responses that may affect the course of periodontitis.

Some of the most upregulated and downregulated genes are worthy of attention. *CXCL5* was among the top 15 upregulated DEGs, while *LCE2* and *LCE3* were among the top 15 downregulated DEGs observed in T2DM-related periodontitis. CXCL5 is a chemokine ligand of the IL-8 receptor type 2 that has proinflammatory activities, including chemotaxis and angiogenesis [15, 16]. In support of our findings, previous studies observed an association between increased CXCL5 levels and DM complications [16, 17]. Noteworthy, lipopolysaccharide (LPS) from Porphyromonas gingivalis, in combination with advanced glycation end products (AGE), induced greater CXCL5 expressions from monocyte and oral keratinocyte cells [18]. Although the biological functions of LCE are not entirely understood, they seem to have antibacterial activities and a role in the epithelial physical barrier [19, 20]. The lack of LCE3B/C causes dysbiotic changes in the oral and cutaneous microbiota in patients with psoriasis [20]. Other possibly relevant downregulated genes to the pathogenesis of T2DM-related

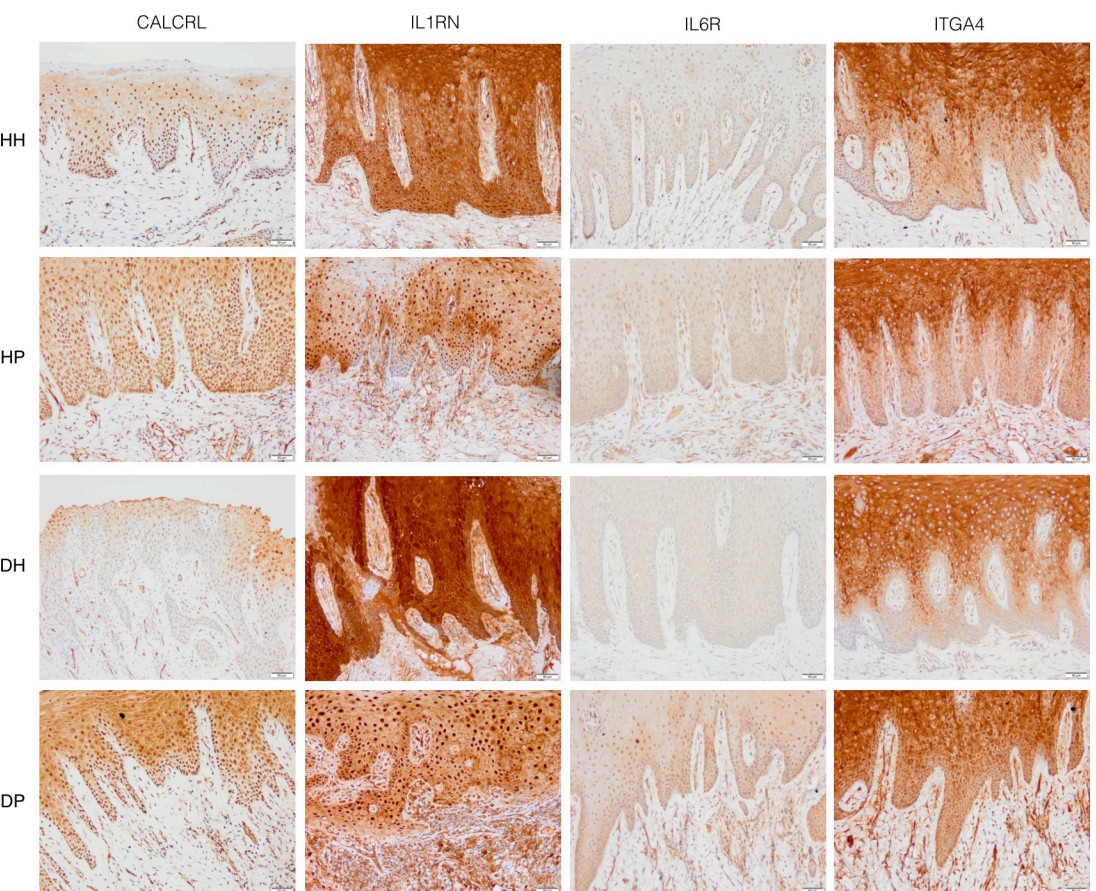

**Fig 4. The immunohistochemical staining using antibodies against CALCRL, IL1RN, IL6R, and ITGA4 in representative gingival biopsies from the HH, HP, DH, and DP groups.** Bar length = 50 μm, original magnification × 20. HH: non-diabetic patients without periodontitis; HP: non-diabetic patients with periodontitis; DH: T2DM patients without periodontitis; DP: T2DM patients with periodontitis.

periodontitis are keratin 76, critical for epidermal integrity [21], and the *PLA2G2F*, as PLA2s are involved in host defense and barrier function [22].

The IPA of the DEGs that were exclusively expressed in periodontitis in diabetic patients revealed a significant upregulation of the Phagosome Formation pathway through the alterations in the expression of genes required for actin cytoskeleton rearrangements, membrane ruffle formation, and phagosome closure (e.g., complement, complement receptor (CR), ITGs, GPCRs, scavenger receptor, toll-like receptor, LPS binding protein, Fc receptor, C-type lectin domain family 4 member E, chemokine receptor, and PL) [23]. Noteworthily, some of the phagosome formation-associated genes that were upregulated in T2DM-related periodontitis (e.g., *C3a*, *C5ar1*, and *LYN*) were also found to be upregulated in other DM complications [24, 25], functioning against microbial invasion and as proinflammatory factors and regulators of immunometabolism [25].

Despite the significant activation of the Phagosome Formation pathway in T2DM-related periodontitis, some critical pathways in response to infections may be impaired in diabetic hosts, including the completion of phagocytosis [26, 27]. Hyperglycemia/DM can affect the expression of complement-related factors required for bacterial opsonization and internalization [26], and reduce the phagosome maturation which succeeds phagosome formation [27]. Finally, the dysregulation of PI3Ks, herein represented by *PIK3R2*, may also affect

phagocytosis [28]. Thus, despite the effort to recruit immune cells and form phagosomes in disease tissues in patients with T2DM, under hyperglycemia, microorganisms may evade phagosome maturation and degradation in the phagolysosomes, leading to persistent infection and inflammation. The impact of DM on different stages of phagocytosis represents an interesting area for further research in the context of periodontitis.

DEGs uniquely expressed in T2DM-related periodontitis were related to the activation of the PI3K/AKT signaling pathway that is activated through various stimuli (e.g., cytokines, growth factors, cellular stresses, and insulin) and regulates several biological processes (e.g., cellular survival, apoptosis, autophagy, proliferation, growth, motility, transcription, and translation) involved in periodontitis and other diseases [29, 30]. Additionally, the PI3K/AKT pathway is reported to regulate insulin and hyperglycemia, and the insulin resistance [29] and the AGE-RAGE axis can activate PI3K/AKT signaling [31]. Finally, the activation of the PI3K/AKT signaling pathway has been associated with DM-related complications [32] and with osteoclastogenesis and bone resorption, an important hallmark of periodontitis [33].

The relevant genes that were involved in the PI3K/AKT pathway activation observed in the T2DM-related periodontitis included those that affect apoptosis (e.g., *FOXO1*, *BCL2*, and growth differentiation factor 15 [*GDF15*]) and cell cycle passage (e.g., *CDKN1B* and *CCND1*). Also abundant were the dysregulated expression of pro- and anti-inflammatory cytokine receptors (e.g., IL18RAP, IL6R, IL12RB2, IL17RC, IL17RD, IL20RA, IL20RB, IL22RA1, IL4R, and IL9R), supporting the notion that cytokines play pivotal roles in the pathogenesis of T2DM-related periodontitis, a typical infectious-inflammatory disorder. The concurrent increased expression of IL-6R at the protein level observed in diseased tissues in diabetic patients indicates the role of the pro-inflammatory IL-6/IL-6R axis in disease pathogenesis. ITGs, which mediate cell-matrix interactions and leukocyte migration and adhesion [34], were consistently overexpressed in T2DM-related periodontitis, and the ITGA4 upregulation was confirmed at the protein level. T2DM-related periodontitis also exhibited marked upregulation of GDF15 (also known as macrophage inhibitory cytokine-1) (Table 1), which was reported to play a role in regulating inflammatory and apoptotic pathways and whose expression was found to be significantly higher in multiple tissues of diabetic patients [35]. Finally, the downregulation of kinases and phosphatase-related genes in diseased tissues in diabetic patients is consistent with the role of the PI3K/AKT signaling as a classical phosphorylation cascade, where the protein kinases regulate phosphorylation and the phosphatases (e.g., PPP family) control dephosphorylation reactions. Remarkably, *PIK3R2*, a PI3K subunit involved in multiple canonical pathways, was downregulated at the mRNA level but was upregulated at the protein level in T2DM-related periodontitis. This downregulation could be due to the low stability of mRNA and/or the reduced proteasomal degradation of PIK3R2 since PTPN13, which dephosphorylates PIK3R2 at Tyr-655 [36], was decreased at the mRNA and protein levels. However, further studies are needed to confirm this assumption.

The IPA also predicted the activation of the Cardiac β-adrenergic and tRNA Splicing pathways in the T2DM-related periodontitis, which were predominantly driven by the upregulation of phosphodiesterases (PDE 1, 4, 5, 7, and 10), a superfamily of cAMP/cGMP hydrolyzing enzymes. The upregulation of PDEs has been linked to the cellular decrease of cAMP and cGMP and to the increase of oxidative stress and pro-inflammatory factors. Hence, the dysregulation of PDEs has been associated with inflammation, DM, DM complications, and imbalanced bone homeostasis [37] whereas the inhibition of PDEs has been suggested as a strategy for treating infectious and inflammatory conditions [37–39]. Thus, the use of PDE inhibitors as an add-on treatment for T2DM-related periodontitis seems to be an attractive research topic.

The inactivation of the ADM signaling pathway was predicted in T2DM-related periodontitis. The downregulated genes associated with the ADM pathway included MAPKs, PLCs, the

PIK3R2, the guanylate cyclase C, implicated in infection and inflammation control and the IL1RN, which encodes the anti-inflammatory protein IL-1Ra. Worthy of note, IL1RN mRNA downregulation was validated at the protein level by IHC. The ADM signaling pathway was also related to the upregulation of BCL2 and caspase-3, regulators of apoptosis, C3, an activator of the complement system, and IL1A, a pro-inflammatory mediator. Importantly, ADM has antimicrobial properties in the oral mucosal epithelia [40] and acts as an angiogenic, anti-apoptotic, and anti-inflammatory factor through the G protein-coupled receptor 182 (GPR182) and the receptor activity modifying protein 2 (RAMP2)/CALCRL complex [41]. While *ADM2* was upregulated in periodontitis in the diabetic and non-diabetic patients (S4 Table), *GPR182*, *CALCRL* (validated at the protein level), and *RAMP2* were significantly upregulated in the diabetic patients only, suggesting that ADM may perform its biological functions mainly in T2DM-related periodontitis. This hypothesis needs to be tested in further studies.

The superpathway of Cholesterol Biosynthesis and some of its related DEGs were downregulated in the T2DM-related periodontitis, suggesting the existence of a transcriptional regulation connecting this metabolic pathway with the immune system. DM and insulin deficiency have been reported to suppress the production of cholesterol synthesis enzymes and the transcriptional regulator sterol regulatory element-binding protein-2 (*SREBF2*; here, downregulated in the diabetic patients only; S3 Table) [42]. Overall, the reduction of cholesterol at a cellular level and the dysregulation of its precursors might affect cell proliferation, cycle progression, apoptosis, inflammation, and phagocytosis [43]. Notably, cholesterol intermediates may play biological functions besides cholesterol biosynthesis, which may affect inflammation and immunity [44]. Dehydrocholesterol reductase (*DHCR*) 7, for example, is involved in vitamin D synthesis, macrophage polarization, and regulation of IL-10 expression. Transmembrane 7 superfamily member 2 (*TM7SF2*) acts on the cell cycle, differentiation, and apoptosis and directs an anti-inflammatory loop [43, 44]. DHCR24, here, downregulated at the mRNA and protein levels in the T2DM-related periodontitis, has antiapoptotic functions and inhibits the generation of intracellular reactive oxygen species [43, 44] and LPS-induced osteoclastogenesis [45]. Future in-depth studies are required to identify the actual role of cholesterol biosynthesis-related genes on the pathogenesis of T2DM-related periodontitis.

The inhibition of SIPC was predicted in the T2DM-related periodontitis. Inositol phosphates are membrane components that have functions in cell growth, differentiation, migration, apoptosis, and endocytosis. Inositols are considered possible mediators of insulin signaling, and their insufficiency may contribute to insulin resistance, T2DM, and DM complications [46]. Among the proteins encoded by the downregulated DEGs associated with SIPC observed in the T2DM-related periodontitis were predominantly kinases and phosphatases, which not only control phosphorylation but also have further biological functions that can affect periodontitis progression. Dual-specificity phosphatase 14 (*DUSP14*) and inositol 1,4,5-trisphosphate 3-kinase C (*ITPKC*), for example, are both negative regulators of T-cell activation [47]. DUSP14 also has inhibitory effects on inflammation and osteoclastogenesis [45]. PTPN13, downregulated in diseased tissues in the diabetic patients at the mRNA and protein levels, is involved in apoptosis, cell migration, epithelial cell junction stabilization, and epithelial barrier regulation [48]. PTPN2, a vitamin-D-responsive tyrosine-specific phosphatase, acts as a protection factor controlling glucose homeostasis and pro-inflammatory conditions [48]. Therefore, the transcript-level and protein-level overexpression of PTPN2 observed in this study may represent an attempt to control the periodontal breakdown in the diabetic patients. Lastly, the observed downregulation of PLCs may also have implications for the pathogenesis of T2DM-related periodontitis. For instance, the lack or reduction of *PLCD1* and *PLCD3* was associated with increased cell apoptosis and the upregulation of LPS-induced proinflammation [49].

The IPA on the DEGs in the non-diabetic patients revealed less of an impact on the canonical pathways (there were no pathways with a Z-score >2 or <−2). Among the dysregulated genes associated with the Tumor Microenvironment pathway observed in the non-diabetic patients were those that encoded MMPs, which play well-known roles in inflammation and immunity [50]. Importantly, *ATF3*, *FOS*, and *JUN*, that function in Tumor Microenvironment and PI3K Signaling in B Lymphocytes pathways, were exclusively downregulated in periodontitis in the non-diabetic patients. *JUN*, *FOS*, *ATF*, and *FOSB* (among the top ten downregulated genes in the non-diabetic patients; Table 1) are members of the activator protein-1 (AP-1) transcription factor complex, which is involved in cell proliferation, migration, apoptosis, inflammation, and bone remodeling [51]. The analysis of the transcriptomic datasets of human periodontitis (GSE16134 and GSE10334) and the immunosuppression genes by deep learning-based autoencoder techniques suggested that FOS and JUN were downregulated transcription factors with possible roles in periodontitis [52]. Using the same transcriptomic database (GESE16134), FOSB was identified as an upregulated DEG with potential participation in periodontitis [53]. Future studies are needed to explore the actual involvement of the AP-1 complex-related genes in periodontitis.

Long non-coding RNAs (lncRNAs) may have coding potential and affect the transcription, post-transcription, and translation of other genes and the activation or suppression of various encoding molecules and signaling pathways [54–57]. In this study, tissues with periodontitis in diabetic and non-diabetic patients exhibited dysregulation in the expression of several lncRNAs, supporting the findings of previous investigations [57, 58]. lncRNAs were also suggested to play critical roles in DM complications [59, 60]. Intriguingly, two lncRNAs ranked in the top 15 downregulated DEGs (SCARNA2 and SMAD1-AS1) were only expressed in the healthy tissues in diabetic patients (Table 1). SCARNA2 has been reported to participate in DNA repair [61], whereas the biological functions of SMAD1-AS1 are still unknown. To understand the roles of lncRNAs in the pathogenesis of T2DM-related periodontitis is a future challenge.

A strength of this study is to be the first RNA-seq experiment evaluating the exclusively DEGs in T2DM-related periodontitis. Moreover, data analyses were expanded by using IPA to distinguish the associated signaling pathways. Furthermore, a valuable database for the future functional validation of DEGs and signaling pathways was generated.. Although a previous study indorsed a minimum of five samples per group for RNA-Seq datasets [62], one of the limitations of the present study is the small sample size. Hence our findings should be interpreted with caution and may need to be confirmed in a large-scale study. Another limitation is the analysis of the entire gingival transcriptomic profiles from a mix of cells that tends to reflect the gene expression patterns of predominant cell types (e.g., epithelial cells and fibroblasts). Thus, additional studies are needed to elucidate individual cell transcription profile involved in T2DM-related periodontitis. Furthermore, our study focused on uncontrolled T2DM. Future studies should also evaluate the effects of well-controlled T2DM on the transcriptomic profile of periodontium. Also, the cross-sectional nature of this study does not allow for causal and temporal inferences on the effects of a given gene/pathway on disease progression. Finally, to obtain a comprehensive understanding of the biological system underling the pathogenesis of the T2DM-related periodontitis, additional omics-based methodologies (e.g., proteomics) should be considered in future studies.

## Conclusion

*CXCL5* was one of the most upregulated genes, while *LCE2* and *LCE3* were considerably downregulated T2DM-related periodontitis. Furthermore, T2DM-related periodontitis was

associated with the dysregulation of functional pathways, including the activation of Phago-some Formation, Cardiac β-adrenergic, tRNA Splicing, and PI3K/AKT pathways and inhibition of Cholesterol Biosynthesis, Adrenomedullin, and Inositol Phosphate Compounds pathways. The idenfication of unique transcriptomic signatures can be used as potential targets for the development of future diagnostic, preventive and therapeutic approaches.

## Supporting information

**S1 Fig. The DEGs expressed in the non-diabetic patients (yellow), in patients with T2DM (blue) and the commonly expressed in both the non-diabetic and diabetic patients.** HH: non-diabetic patients without periodontitis; HP: non-diabetic patients with periodontitis; DH: T2DM patients without periodontitis; DP: T2DM patients with periodontitis.
(JPEG)

**S1 Table. Demographic characteristics, glycemic and periodontal parameters (mean ± SD) of the study population.**
(DOCX)

**S2 Table. DEGs in tissues with periodontitis in non-diabetics relative to healthy tissues in non-diabetics exclusively.**
(XLSX)

**S3 Table. DEGs in tissues with periodontitis relative to healthy tissues observed in patients with T2DM exclusively.**
(XLSX)

**S4 Table. DEGs in tissues with periodontitis in relation to healthy tissues observed in non-diabetic patients and in patients with T2DM concurrently.**
(XLSX)

**S5 Table. Ingenuity canonical pathways related to DEGs expressed exclusively in non-diabetic patients and the respective genes that function within each of these signaling pathways.**
(DOCX)

**S6 Table. Ingenuity canonical pathways related to the DEGs expressed exclusively in patients with T2DM and the respective genes that function within these signaling pathways.**
(DOCX)

## Acknowledgments

We thank Dr. Ann Fu Dongtao for her technical assistance with immunohistochemistry.

## Author Contributions

**Conceptualization:** Poliana Mendes Duarte, Ikramuddin Aukhil.

**Data curation:** Tamires Szeremeske Miranda, Juliana Sardenberg.

**Formal analysis:** Tongjun Gu.

**Funding acquisition:** Poliana Mendes Duarte.

**Investigation:** Poliana Mendes Duarte, Bruno César de Vasconcelos Gurgel, Tamires Szeremeske Miranda, Ikramuddin Aukhil.

**Methodology:** Poliana Mendes Duarte, Bruno César de Vasconcelos Gurgel, Tamires Szeremeske Miranda, Juliana Sardenberg, Ikramuddin Aukhil.

**Supervision:** Poliana Mendes Duarte.

**Validation:** Poliana Mendes Duarte, Bruno César de Vasconcelos Gurgel, Ikramuddin Aukhil.

**Writing – original draft:** Poliana Mendes Duarte, Bruno César de Vasconcelos Gurgel.

**Writing – review & editing:** Poliana Mendes Duarte, Bruno César de Vasconcelos Gurgel, Tamires Szeremeske Miranda, Ikramuddin Aukhil.

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
