## [Decision Letter · Decision Letter 0]

7 Nov 2023

PONE-D-23-19661Distinctive genes and signaling pathways associated with type 2 diabetes-related periodontitis: a preliminary studyPLOS ONE

Dear Dr. Duarte,

Thank you for submitting your manuscript to PLOS ONE. After careful consideration, we feel that it has merit but does not fully meet PLOS ONE’s publication criteria as it currently stands. Therefore, we invite you to submit a revised version of the manuscript that addresses the points raised during the review process.

As you may find in the reviewers comments the manuscript should be changed throughly before it may be considered for publication. The conclusions should be better explained and the limitations of the study should be stressed. Besides the methodological suggestions of the reviewr 2should be followed.

We look forward to receiving your revised manuscript.

Kind regards,

Víctor Sánchez-Margalet

Academic Editor

PLOS ONE

Journal Requirements:

"All funding was provided by University of Florida College of Dentistry Faculty Seed Grant. We thank Dr. Ann Fu Dongtao for her technical assistance with immunohistochemistry."

"PMD received a University of Florida College of Dentistry Faculty grant. The funders had no role in study design, data collection and analysis, decision to publish, or preparation of the manuscript."

Reviewers' comments:

Reviewer's Responses to Questions

**Comments to the Author**

1. Is the manuscript technically sound, and do the data support the conclusions?

Reviewer #1: No

Reviewer #2: Yes

Reviewer #3: Yes

2. Has the statistical analysis been performed appropriately and rigorously? 

Reviewer #1: No

Reviewer #2: Yes

Reviewer #3: Yes

3. Have the authors made all data underlying the findings in their manuscript fully available?

Reviewer #1: No

Reviewer #2: Yes

Reviewer #3: Yes

4. Is the manuscript presented in an intelligible fashion and written in standard English?

Reviewer #1: Yes

Reviewer #2: Yes

Reviewer #3: Yes

5. Review Comments to the Author

Reviewer #1: The biological mechanisms underlying the pathogenesis of type 2 diabetes (T2DM) related periodontitis remain unclear. This cross-sectional study evaluated the distinctive transcriptomic changes between tissues with periodontal health and with periodontitis in patients with T2DM. In this cross-sectional study, whole transcriptome sequencing was performed on gingival biopsies from non-periodontitis and periodontitis tissues from non-diabetic and diabetic patients. A differentially expressed gene (DEG) analysis and Ingenuity Pathway Analysis (IPA) assessed the genes and signaling pathways associated with T2DM-related periodontitis.

It is preliminary study. It may be submitted again after completion of the project.

Reviewer #2: This study evaluated the distinctive transcriptomic changes between tissues with periodontal health and with periodontitis in patients with T2DM, through the analysis of the transcriptome performed on gingival biopsies from non-periodontitis and periodontitis tissues from non-diabetic and diabetic patients.

The manuscript is well written, the study well designed and the discussion of the findings was carried out in a detailed and precise manner.

I made some comments that can be found in the attached file.

Reviewer #3: Dear Authors,

I appreciate your time and effort in conducting this study.

In this study, the authors evaluated the whole transcriptome profile and related histopathological analysis of the periodontitis patients with and without T2DM and non-periodontitis patients with and without T2DM from gingival tissue samples.

The determination of the numbers of study groups, collection of samples during which surgical procedures, how the samples for IHC were stored, the reason why 4 samples were selected from each group for IHC, and why proteomics was not conducted have not been explained.

I believe that the results of this study necessitate researchers to write a more specific conclusion. What is expected to be served by the results of this study?

I hope I could provide some support.

Kind Regards

6. PLOS authors have the option to publish the peer review history of their article (what does this mean?). If published, this will include your full peer review and any attached files.

Reviewer #1: No

Reviewer #2: No

Reviewer #3: **Yes: **Esra Guzeldemir Akcakanat

---

## [Author Response · Author response to Decision Letter 0]

27 Nov 2023

AUTHOR’S RESPONSE

We thank the Associate Editor and the Referees for their time and for the constructive comments regarding this manuscript. Please, find below the point-by-point answers to the Reviewers' comments.

Reviewer’s comments: 1. Please ensure that your manuscript meets PLOS ONE's style requirements, including those for file naming. The PLOS ONE style templates can be found at 

Author’s response: PLOS ONE's style requirements were revised accordingly. 

Reviewer’s comments: 2. Thank you for stating the following in the Acknowledgments Section of your manuscript: "All funding was provided by University of Florida College of Dentistry Faculty Seed Grant. We thank Dr. Ann Fu Dongtao for her technical assistance with immunohistochemistry." We note that you have provided funding information that is not currently declared in your Funding Statement. However, funding information should not appear in the Acknowledgments section or other areas of your manuscript. We will only publish funding information present in the Funding Statement section of the online submission form. 

"PMD received a University of Florida College of Dentistry Faculty grant. The funders had no role in study design, data collection and analysis, decision to publish, or preparation of the manuscript." Please include your amended statements within your cover letter; we will change the online submission form on your behalf.

Author’s response: Acknowledgments Section was revised accordingly. Please, keep the same funding statement. 

Reviewer’s comments: 3. Your ethics statement should only appear in the Methods section of your manuscript. If your ethics statement is written in any section besides the Methods, please delete it from any other section. 

Author’s response: Ethics statement was revised accordingly. It is now only in the Methods section. 

Review Comments to the Author

Reviewer #1: 

Reviewer’s comments: The biological mechanisms underlying the pathogenesis of type 2 diabetes (T2DM) related periodontitis remain unclear. This cross-sectional study evaluated the distinctive transcriptomic changes between tissues with periodontal health and with periodontitis in patients with T2DM. In this cross-sectional study, whole transcriptome sequencing was performed on gingival biopsies from non-periodontitis and periodontitis tissues from non-diabetic and diabetic patients. A differentially expressed gene (DEG) analysis and Ingenuity Pathway Analysis (IPA) assessed the genes and signaling pathways associated with T2DM-related periodontitis. It is preliminary study. It may be submitted again after completion of the project.

Author’s response: Thank you so much for taking the time to review our manuscript. We classified our study as preliminary because of its exploratory nature and sample size (please, note that the study was performed using a University of Florida Seed Grant). However, considering the magnitude and richness of the data generated, we believe that the study is certainly powered to provide relevant information to the scientific community. Based on these initial validated data, we are going to definitely continue to investigate the potential pathways and mechanisms involved in the pathogenesis of type 2 DM-related periodontitis by applying additional methodologies and a larger population. However, we believe that data generated in this initial study is important and valuable and can contribute to other research groups that investigate the pathogenesis of periodontitis in patient with diabetes. Thus, we humbly would like to ask this reviewer to consider the publication of these findings as they are. 

Reviewer #2

Reviewer’s comments: The authors presented the periodontal parameters with mean and SD. The authors could stratify the sample regarding the periodontal parameters to better characterize the sample, especially for CAL and PD. For example: 2-3 mm, 4-5 mm and greater than 6 mm.

Author’s response: Information about the full-mouth percentage of sites with different PD (PD ≤ 3mm, PD=4-6mm and PD ≥ 7 mm), and clinical attachment loss (no clinical attachment loss and with CA loss of 1-2 mm, 3-4mm and ≥ 5 mm) categories was added in S1 table, as requested by this reviewer. Please, note that some periodontally healthy patients could have sites with PD > 3mm, assuming pseudopockets. Furthermore, they could also present some sites with CAL, but those were non-interdental and attributed to non-periodontitis reasons. This information was taken from patients records.

Reviewer’s comments: Writing gene names in humans should be written in italics.

Author’s response: Gene names were revised accordingly. 

Reviewer’s comments: Have you ever think about GSEA (Gene Set Enrichment Analysis)? I believe that the statistical approach of GSEA strengthened the results regarding the gene sets involved in the biological process of the diseases studied here.

Author’s response: Considering the exploratory nature of this study, we chose the Ingenuity pathway analysis (IPA, QIAGEN) to initially explore our transcriptomic data and to provide information about pathways, genes and other signatures that may be significantly altered across our groups. The authors are aware that GSEA employs gene sets and features that have been assumed to be associated with a given disease or pathway in order to suggest biological applications. Thus, the GSEA analysis suggested by this Reviewer is a valuable complementary/confirmatory analysis that our group intends to apply in future studies. However, at this stage, we would need to start the study from the very beginning, doing new bioinformatics, results presentation (tables and figures), data interpretation and discussion. So, given the impracticality of carrying out the suggested analysis at this moment, we would be grateful if this reviewer agrees that the IPA analyses presented in our preliminary study are sufficient and provide relevant information that may guide future studies in the field. 

Reviewer’s comments: The authors could provide more information about their findings in the conclusion section, drawing attention to the main pathways found and some differently expressed genes.

Author’s response: Conclusion section was revised as suggest by this reviewer and reviewer #3.

Reviewer’s comments: I suggest reporting the study methodology according to STROBE guidelines.

Author’s response: Thank you very much for this suggestion. STROBE guidelines were adopted for study conducting and reporting findings 

Reviewer’s comments: How did the authors set the sample size?

Author’s response: This reviewer raised an important question about our study sample size. Studies using bulk RNA sequencing in human gingival tissues with periodontitis varies considerably in terms of sample size (from 6 to more than 10 sample per group) (e.g. Chen et al. doi: 10.1007/s00784-023-05017-y, Oh et al. doi: 10.1002/JPER.23-0289, Kim et al. doi: 10.1186/s40246-016-0084-0). However, so far, no previous study performed RNA sequencing of human gingival samples with periodontitis and diabetes to be used as a reference for our sample size calculation. Thus, as we did not know the actual difference in the means and variance of genes that are significantly changed between our case and control samples from prior studies, a precise sample size could not be calculated. Considering the exploratory characteristic of our study (preliminary study), sample size was established based on previous recommendation of a minimum of five samples per group for RNA-Seq datasets (Ching et al. 2014). It is important to highlight that we are aware that sample size is a limitation of our study and that our findings should be confirmed in a large-scale study. This is stated in the discussion section (last paragraph). 

Reviewer’s comments: Who performed the diagnosis of the periodontal condition? Was it one or more examiners? Were the examiners who performed the periodontal examination calibrated to each other?

Author’s response: Our transcriptome analysis focused on waste gingival tissues from specific teeth not on patient’s full-mouth periodontal condition. Full-mouth data are presented in S1 Table in order to characterize the severity of periodontitis of the study population. Thus, full-mouth clinical examination was not performed by a single examiner but obtained from patient’s records. However, in order to have accuracy in the periodontal diagnosis of the sampled teeth, periodontal parameters (i.e., probing depth, clinical attachment level, bleeding on probing, furcation involvement and mobility) were recorded around the specific sampled teeth by the same trained and calibrated examiner (P.M.D). The study examiner participated in a previous calibration exercise, and the standard error of measurement was estimated. Intra-examiner variability was recorded as 0.20 mm for PD and 0.23 mm for CAL. This information was included in the Material and Methods section – Population. 

Reviewer’s comments: I believe that a group of patients with well-controlled DM could bring important information about the influence of DM on the pathogenesis of Periodontitis. The absence of this group does not make the results obtained unfeasible, but a discussion about this would be interesting.

Author’s response: Thank you very much for this suggestion. At this moment, we only focused on uncontrolled T2DM because we assumed that it would affect more the pathogenesis of periodontitis. But we agree with this reviewer that a group of well-controlled patients would be very interesting. Thus, it will be considered in future studies. This consideration was included in the discussion section, as suggested by this reviewer (Please, see Discussion Section – last paragraph). 

Reviewer’s comments: Do the authors intend to deposit the transcriptome data in a public domain database, such as Gene Expression Omnibus (GEO)?

Author’s response: According to our data availability statement, all data are available in the main text or in the supplementary materials. We do not intend to deposit it in a public domain database at this time. 

Reviewer #3

Reviewer’s comments: This study evaluated the distinctive transcriptomic changes between tissues with periodontal health and with periodontitis in patients with T2DM, through the analysis of the transcriptome performed on gingival biopsies from non-periodontitis and periodontitis tissues from non-diabetic and diabetic patients. The manuscript is well written, the study well designed, and the discussion of the findings was carried out in a detailed and precise manner. I made some comments that can be found in the attached file.

Author’s response: Thank you so much for the encouraging comments about our manuscript.

Reviewer’s comments: The determination of the numbers of study groups, collection of samples during which surgical procedures, how the samples for IHC were stored, the reason why 4 samples were selected from each group for IHC, and why proteomics was not conducted have not been explained. 

Author’s response: All explanations requested by this reviewer was included in the Material and Methods section (please, see tack changes). 

Determination of the number of study groups: Considering that we would like to observe the differently expressed genes and pathways in patients with periodontitis and in patients with DM, we had to use periodontally-healthy patients and patients without DM as references for the transcriptome analysis and IHC. Thus, “based on the periodontal and DM conditions, patients were assigned into one of the following four groups: non-diabetic patients without periodontitis (HH; n=9), non-diabetic patients with periodontitis (HP; n=9), T2DM patients without periodontitis (DH; n=9), and T2DM patients with periodontitis (DP: n=9).” (Please, see Material and Methods - Population). 

Collection of samples during which surgical procedures: Collection of samples was detailed, as requested by this reviewer (Please, see Material and Methods – Tissue collection). 

How the samples for IHC were stored: Thank you for this observation. The method used for sample fixation and storage was described accordingly (Please, see Material and Methods – IHC). 

The reason why 4 samples were selected from each group for IHC: In this study, IHC was performed for a total of eight antibodies in four different groups, totalizing 128 sections. At this exploratory phase, we sought to present only descriptive analyses of the IHC sections, which does not require a large number of repetitions. Thus, sample size calculation was not performed. Considering the investigative characteristic of our study, we thought that the descriptive method would give us valuable details (staining location in terms of tissue [epithelium, connective tissue, or vessels], cell type and cell site [membrane, cytoplasm or nuclear], which may be hidden by quantitative methods, using scoring system categorization. 

Why proteomics was not conducted: This reviewer raised an important question about the performance of additional omics analyses. Unfortunately, due to a budget constraint, at this moment, our analysis limited to transcriptome analysis, followed by validation of the main results at the protein level by IHC. Additional omics analyses will be certainly considered in future studies. This was stated in the discussion section. (Please, see Discussion Section – last paragraph). 

Reviewer’s comments: I believe that the results of this study necessitate researchers to write a more specific conclusion. What is expected to be served by the results of this study?

Author’s response: Conclusion section was revised, as suggested by this reviewer and reviewer #2.

---

## [Decision Letter · Decision Letter 1]

26 Dec 2023

Distinctive genes and signaling pathways associated with type 2 diabetes-related periodontitis: a preliminary study

PONE-D-23-19661R1

Dear Dr. Mendes Duarte

We’re pleased to inform you that your manuscript has been judged scientifically suitable for publication and will be formally accepted for publication once it meets all outstanding technical requirements.

Kind regards,

Víctor Sánchez-Margalet

Academic Editor

PLOS ONE

Additional Editor Comments (optional):

Reviewers' comments:

Reviewer's Responses to Questions

**Comments to the Author**

1. If the authors have adequately addressed your comments raised in a previous round of review and you feel that this manuscript is now acceptable for publication, you may indicate that here to bypass the “Comments to the Author” section, enter your conflict of interest statement in the “Confidential to Editor” section, and submit your "Accept" recommendation.

Reviewer #2: All comments have been addressed

Reviewer #3: All comments have been addressed

2. Is the manuscript technically sound, and do the data support the conclusions?

Reviewer #2: Yes

Reviewer #3: Yes

3. Has the statistical analysis been performed appropriately and rigorously? 

Reviewer #2: Yes

Reviewer #3: Yes

4. Have the authors made all data underlying the findings in their manuscript fully available?

Reviewer #2: Yes

Reviewer #3: Yes

5. Is the manuscript presented in an intelligible fashion and written in standard English?

Reviewer #2: Yes

Reviewer #3: Yes

6. Review Comments to the Author

Reviewer #2: I would like to congratulate the authors for their effort in properly reviewing the manuscript. I believe that with the reviewers' suggestions the study became more robust and accurate.

Reviewer #3: (No Response)

7. PLOS authors have the option to publish the peer review history of their article (what does this mean?). If published, this will include your full peer review and any attached files.

Reviewer #2: No

Reviewer #3: No

---

## [Editor Report · Acceptance letter]

10 Jan 2024

PONE-D-23-19661R1 

PLOS ONE

Dear Dr. Duarte, 

I'm pleased to inform you that your manuscript has been deemed suitable for publication in PLOS ONE. Congratulations! Your manuscript is now being handed over to our production team.

Kind regards, 

on behalf of

Dr. Víctor Sánchez-Margalet 

Academic Editor

PLOS ONE